# Research on the coordination mechanism of major industrial project engineering and construction multi-agents based on structural holes theory

Xiaokang Han ⃝*, Wenzhou Yan, Mei Lu

School of Management, Xi'an University of Architecture and Technology, Xi'an, China

* 83263972@qq.com

**Data Availability Statement:** All relevant data are within the paper and its Supporting Information files.

## Abstract

Industry is an important pillar of the national economy. Industrial projects are the most complex and difficult projects to control in the construction industry, and major industrial projects are even more complex and difficult to control. Multi-agent coordination is one of the core issues of industrial projects. Based on an analysis of the engineering and construction chains and agent relationships and agent networks of industrial projects, a complex network of the engineering and construction agents of industrial projects is established, and the complex network structural holes theory is applied to study the nonrepeated relationships among agents in industrial projects. Assuming agents are linked through contract relations and the most critical contract index between the agents in the contract amount, through structural hole analysis considering the EPC and PMC model, the aggregate constraint list is obtained, 2D network diagram and 3D network diagram are shown. According to the aggregate constraint value, the EPC contractor with the minimum aggregate constraint value and the project management company with the minimum aggregate constraint value are the critical agent in EPC and PMC model. By analyzing the complex network comprising different models of industrial projects, it is concluded that the characteristics of the agent maintain an advantage in competition, the coordination mechanism of the integration of agent interests, and multi-agent relations are considered to solve the multi-agent coordination problem in major industrial projects.

## 1 Introduction

Major projects include infrastructure or plant projects with a large investment scale, high technical complexity and difficult control process, such as transportation, energy, chemical industry, communications, municipal administration and aerospace projects, which exert an important influence on political, economic, social, science and technology, engineering development, environmental protection, public health, national security and strategic planning aspects.

**Funding:** The authors received no specific funding for this work.

**Competing interests:** The authors have declared that no competing interests exist.

With the development of science and technology, industrial projects tend to be increasingly large-scale and intensive projects. Industrial projects are characterized by a large scale, high investment, complex process system, intensive layout, open-air features, automatic control, high construction technology requirements, numerous design institutes, several suppliers, many construction units, various cooperation units, countless work interface relations, and long construction cycles, and the agent coordination problem of industrial engineering and construction projects is difficult to solve, which requires a new analysis method of the essential characteristics.

## 2 Background and hypotheses

Complex networks have been widely applied in many fields, including social, technical, information, biological and economic networks. From the Internet to the Worldwide Web, from the structural network of organisms to the food chain among animals, from the neural network of the human body to the social relationship network, and from the construction chain of engineering to the relationship network of engineering agents, complex networks occur everywhere. The research of complex networks permeates various fields, such as physics, biology, social science and engineering, and the qualitative and quantitative research of complex networks has become a major research topic.

In 1977, Freeman [1] proposed the Betweenees, which measures the role and influence of nodes in the entire network. In 1979 and 1980, Freeman [2, 3] studied the centrality in social networks, discussed centrality through experiments, and clarified the concept of Centrality. In 1992, Burt [4] proposed the structural holes theory based on the theory of interpersonal weak ties. In 1998, Watts and his supervisor, Professor Strogatz [5], published a paper titled Collective Dynamics of Small-World Networks in Nature. In 1999, Professor Barabási and his PhD student Albert [6] published a paper, namely, Emergence of Scaling in Random Networks, in Science. In 2009, Rodan [7] empirically tested a mediated moderation model, which distinguished between the five different theoretical mechanisms, i.e., autonomy, competition, information brokering, opportunity recognition and innovation, and their findings suggested that of these five theoretical causal drivers, innovation plays a key role in linking the network structure and content to performance. In 2012, Latora et al. [8] proposed a new measure to reconcile closed and open structures, Simmelian brokerage, that captures opportunities of brokerage between otherwise disconnected cohesive groups of contacts. In 2012, Phelps et al. [9] developed a comprehensive framework that organizes the knowledge networks literature, which was used to review extant empirical research within and across multiple disciplines and levels of analysis, and identified points of coherence and conflict in theoretical arguments and empirical results within and across levels and identified emerging themes and promising areas for future research. In 2015, Burt [10] studied the strengthening of structural holes and proposed a reinforcement measure. In 2018, Tohyun et al. [11] studied focuses on the joint effects of the firms' access to structural holes within social networks and their status within social hierarchy on their innovation performance, argued that the effects of structural holes and status contradict, rather than complement, each other because one tends to interfere with the other, and found that the positive effects of structural holes tend to be relatively stronger among lower-status firms, whereas the negative effects become stronger as the firms' status increases. In 2019, Deng et al. [12] proposed a new measurement model for critical nodes based on global features and local features, which considers the edge betweenness and edge clustering coefficients and combines the mutual influence between nodes and edges in a network, and proposed an algorithm based on the aforementioned model Subsequently. In 2020, Alizadeh et al. [13] proposed a dynamic DEA (DDEA) model with time-based dependencies

between the successive periods for assessing the performance over successive periods, and found that the efficiencies of power generation and transmission sectors are decreasing while the distribution performance is increasing. Han et al. [14] proposed a reasoning model for emergency measures can be applied in the scheduling control of industrial projects, which is an excellent way to provide effective case support and decision data for the improvement of early warning and feedback tracking theory in project scheduling control. Han [15] proposed a WBS-free scheduling method based on database relational model, which solve the problem of diversity in scheduling form and implement the innovation of scheduling method. Han et al. [16] proposed an improved ant colony algorithm to determine the critical path by setting the path distance and time as negative, while the transition probability remains unchanged.

Since the 21st century, scholars have carried out extensive empirical studies on complex systems in different fields, such as criminal networks, Twitter, community networks, and interpersonal communication, in terms of the complex network centrality and cluster coefficient theory.

## 3 Structural holes theory

Structural holes are the nonrepeated relationships between two agents. The benefits contributed by the two agents associated with the structural holes to the network are accumulative but not overlapping. There are two indexes to measure structural holes: cohesion and structural allelism. If strong ties occur between two agents, repeated agent relations exist that lack structural holes, and cohesion redundancy follows. A structural hole exists if two of neighbours are not linked to each other. Through these two neighbours, they are connected to different parts of the larger network, and thus have access to different sources of dispersed information. Thus if a firm is to form a new link, closing a structural hole is less valuable than finding a partner to whom none of the current partners is currently connected [17]. The argument is that agents attempt to increase their betweenness centrality. Rodan empirically tested a mediated moderation model that distinguishes between the five different theoretical mechanisms: autonomy, competition, information brokering, opportunity recognition and innovativeness, and the findings suggested that of these five theoretical causal motors, innovativeness plays a key role in linking network structure and network content to performance [18]. Xing et al. analyzed the spreading effect of industrial sectors with complex network model under perspective of econophysics, and the industrial complex network based on input-output tables from WIOD was proposed to be a bridge connecting accurate static quantitative analysis and comparable dynamic one [19]. Latora et al. attempted to reconcile closed and open structures by proposing a new measure, Simmelian brokerage, that captures opportunities of brokerage between otherwise disconnected cohesive groups of contacts, and the implications of our findings for research on social capital and complex networks were discussed [20]. Zhao et al. proposed a novel measure based on Structural Holes and Degree Centrality(SHDC) which combined Structural Hole and Degree Centrality to measure the node influence, and the method used Degree Centrality to make a fast and coarse distinction between the influence of nodes and uses Structure Hole to reflect the impact of topological connections among neighbor nodes, which improved the ability to distinguish the influence of nodes in the low time complexity [21].

The advantages of structural holes are that it emphasizes that structural holes in interpersonal networks can bring advantages in information and other resources to organizations and individuals in that position. If there is no direct connection between the two, and the connection must be formed through the third party, then the third party acting occupies a structural hole in the network of the relationship. Obviously, the structural hole is for the third party.

A structural hole is considered to measure the constraint index of networks $p_{ij}$, which is the ratio of the probability value of the relation between nodes $i$ and $j$ to the probability value of all relations of $i$, and $a_{ij}$ is the value of the edge attribute between nodes $i$ and $j$.

$$p_{ij} = \frac{a_{ij} + a_{ji}}{\sum_k (a_{ik} + a_{ki})} \tag{1}$$

Aggregate constraints $c_{ij}$ denote the missing constraints of node $j$ around the initial hole of node $i$.

$$c_{ij} = \left(p_{ij} + \sum_{k,k\neq i,k\neq j} p_{ik}p_{kj}\right)^2 \tag{2}$$

$K$ is the set of all nodes connected to node $i$, for $k \in K$.

The higher the $c_{ij}$ coefficient, the fewer structural holes there are and the higher the network closure is.

The aggregate constraint of node $i$ is $c_i$, corresponding to independent nodes $c_i = 1$.

$$c_i = \sum_j c_{ij} \tag{3}$$

## 4 Industrial project construction stage and agent composition

The industrial project construction stage includes the decision-making, implementation and commercial operation stages, among which the implementation stage includes the engineering, procurement, construction and commissioning stages. Fig 1 shows the industrial project construction stage composition.

The whole construction stage of industrial projects involves the participation of multiple agents who undertake different management tasks and exhibit different interests. Therefore, project management representing agents of different interests is established at the different stages of the project. Fig 2 shows the industrial project construction stage and agent composition.

**Fig 1. Industrial project construction stage composition.**

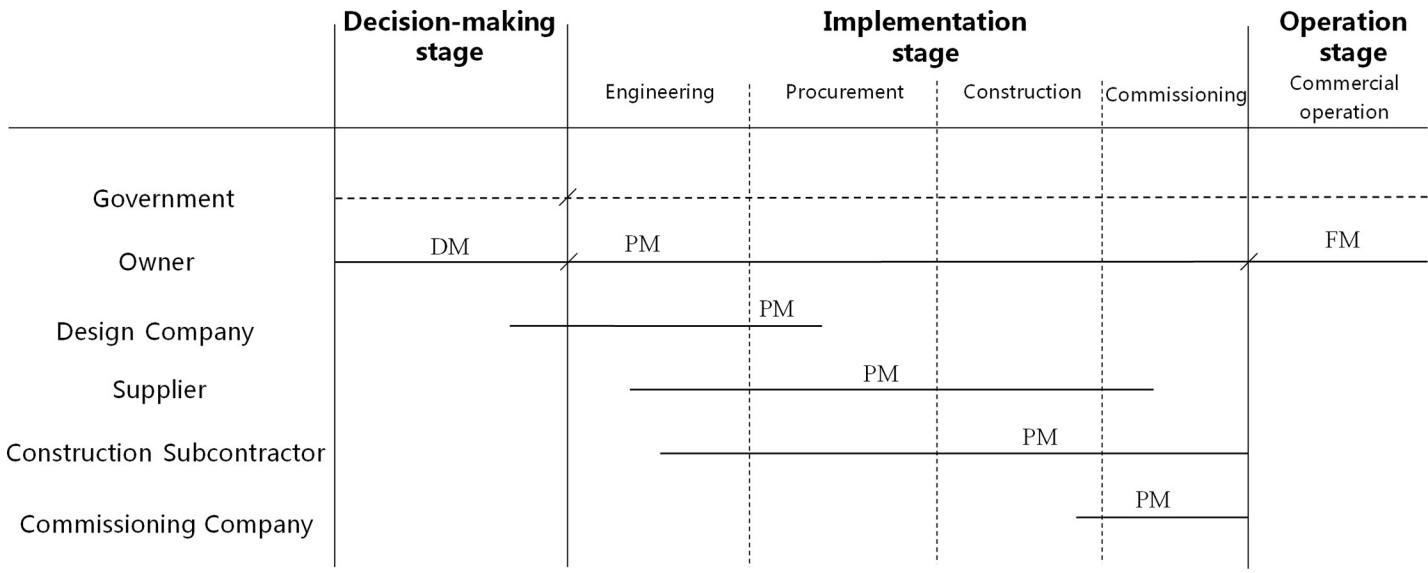

**Fig 2. Industrial project construction stage and agent composition.**

## 5 Engineering and construction chains and agent networks of industrial projects

The engineering and construction chains of industrial projects include the large, complex and systematic industrial projects surrounding the industrial engineering and construction process, from the feasibility study of preparation, process design package, general design, front end engineering design (FEED)/basic design, detailed design, procurement, construction, pre-commissioning, commissioning, start-up, performance testing, final acceptance, and the whole engineering and construction process until the final engineering products are delivered, which includes licensers, design companies, engineering companies, suppliers, and construction subcontractors, to the owner or end user comprise the chain structure as a whole. The engineering and construction chains describe the integration of the interests of multiple agents in the process of industrial project construction, unified allocation of resources between the engineering and construction chains and coordination of the relations among agents to achieve the common interests of all agents. Fig 3 shows a basic model of the engineering and construction chains of industrial projects.

### 5.1 Engineering and construction chain agent relations of industrial projects

The agents of the industrial project engineering and construction chains include owners, supervising companies, project management companies, contractors, licensers, design companies, suppliers, construction subcontractors and commissioning companies.

The multi-agent relationships of the industrial project engineering and construction chains are linked by contracts, and the connecting agents of contracts differ under the various construction models. Under normal circumstances, the principal relationships involving the core or key technology, core or key resources, general supervision and production supervision are contracted by the owner or investor to ensure the overall technology, quality and control of the project. For example, the contract relationship between the licenser, catalyst company, core or key equipment company and supervision company is signed by the owner.

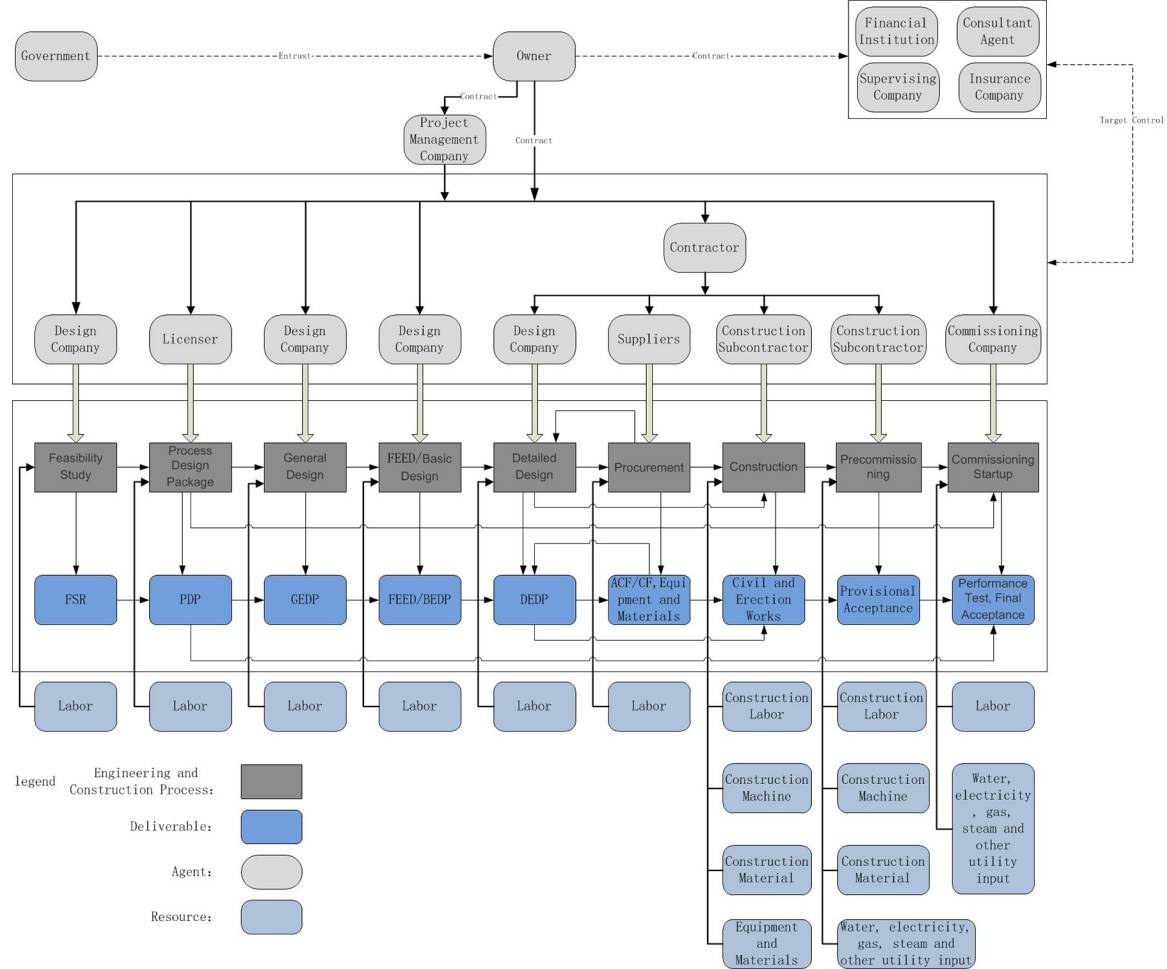

**Fig 3. Basic model of the engineering and construction chains of industrial projects.**

Under the engineering, procurement and construction (EPC) model, the owner establishes a contract with the contractor. All decisions within the work scope of the contract belong to the contractor, while the contractor also assumes all responsibilities within the work scope of the contract, and the contractor establishes contracts with the design company, supplier, construction subcontractor, etc.

Under the project management contractor (PMC) model, the owner establishes a contract with the project management company. The work scope of the project management company is to cooperate with the owner to carry out work without any decision-making power or responsibility. All decision-making power belongs to the owner, who establishes contracts with the design company, supplier, construction subcontractor, etc.

### 5.2 Engineering and construction agent networks of industrial projects

The engineering and construction agent networks of industrial projects are the relationship networks of all agents in the overall process of the engineering and construction chains. Figs 4 and 5 show the Engineering and construction agent networks under the industrial project.

In the Figures above, the arrows of the engineering and construction agent networks of industrial projects only represent the initiator of the subordinate relation, contract or

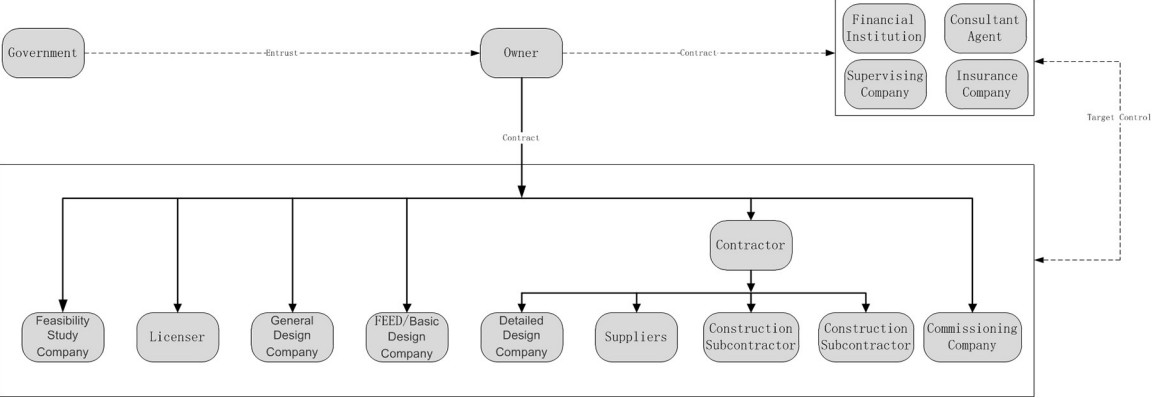

**Fig 4. Engineering and construction agent networks under the industrial project-EPC model.**

entrusted originator, and the relationship between agents is an equal contractual relationship. The flow of information and data is a two-way flow process.

## 6 Engineering and construction agent network analysis of major industrial projects based on complex networks

Industrial projects are characterized by many design companies, several suppliers, numerous construction companies, countless cooperative companies, various work interface relations, and long construction periods, which generally involves hundreds or thousands of agents. They comprise a network of agents through the contract relations and complete the engineering and construction of industrial projects through interaction.

A large number of agents occurs in industrial projects, which constitute a complex agent network that cannot be analyzed with conventional methods. Therefore, it is necessary to analyze the characteristics of the topological structure of the complex network of the agents occurring in industrial projects through software suitable for the analysis and visualization of large networks with thousands or millions of nodes.

Selecting a coal chemical project as an example, the agent network of the industrial project is established, and the network under the different construction models is analyzed.

Overview of the coal chemical project: The project construction scale is the production of 1 million tons/year glycol and 2 million tons/year methanol. The project content mainly

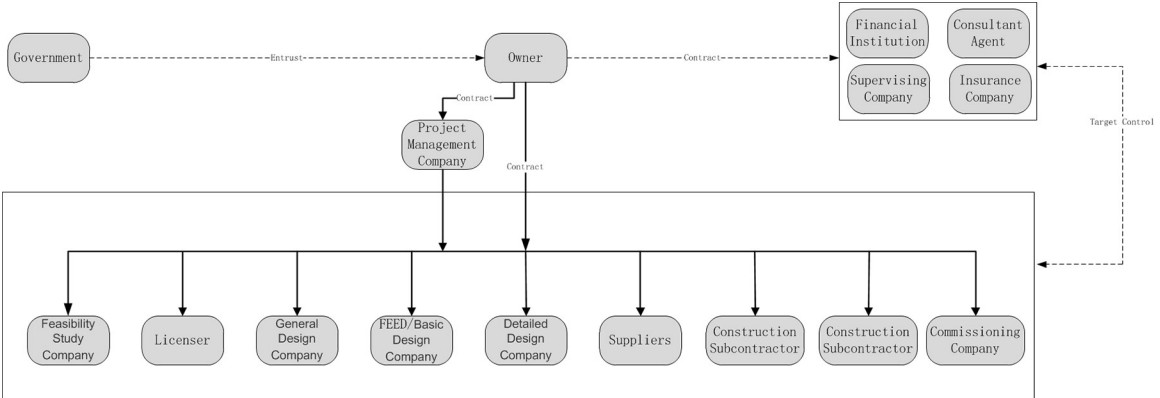

**Fig 5. Engineering and construction agent networks under the industrial project-PMC model.**

includes coal storage and transportation, boiler operation, air separation, gasification, conversion, purification, methanol and ethylene glycol production, tank area maintenance and provision of auxiliary facilities.

The agent relationship of industrial projects is determined by the construction model, and the agent relationship is different under the various construction models. The following analysis of the engineering and construction process includes the agents from the beginning of the project to the final acceptance of the project. The characteristics of the above two construction models, i.e., the EPC and PMC models, are compared through complex network analysis.

Pajek software is adopted to analyze the agent network of major industrial projects. Pajek provides users with a set of fast and efficient algorithms to analyze large-scale (tens of thousands of nodes) complex networks [22]. In Pajek, the time complexity of all algorithms is lower than $O(n^2)$, namely, $O(n)$, $O(n\sqrt{n})$ or $O(n\log n)$. This characteristic of the Pajek algorithm is different from that of other algorithms, and large complex networks are quickly processed, which is also the advantage of Pajek [23].

## 6.1 Complex network analysis of the engineering and construction multi-agents of major industrial projects under the EPC Model

There are a total of 547 agents under the EPC model, as shown in Table 1.

Overview of the coal chemical project: The project construction scale is the production of 1 million tons/year glycol and 2 million tons/year methanol. The project content mainly includes coal storage and transportation, boiler operation, air separation, gasification, conversion, purification, methanol and ethylene glycol production, tank area maintenance and provision of auxiliary facilities.

**6.1.1 Structural hole analysis of the engineering and construction multi-agents of major industrial projects under the EPC model.** The agent network under the EPC model is shown in Fig 6.

Structural hole analysis assigns values $a_{ij}$ to the edges between nodes $i$ and $j$. Agents are linked through contract relations, and the most critical contract index between the agents in the contract amount. Therefore, the contract amount is adopted as the attribute value of the edges between agents to construct a complex network, as shown in Table 2.

Through structural hole analysis considering the EPC model, the aggregate constraint list is obtained, as shown in Table 3, a 2D network diagram is shown in Fig 7, a 3D network diagram is shown in Fig 8, and the 3D network determined through the visualization through similarities (VOS) mapping method is shown in Fig 9.

According to the aggregate constraint value, the owner and EPC contractor attain the minimum aggregate constraint values. Moreover, the 3D diagram of the aggregate constraint analysis of the structural holes shows that the owner and EPC contractor exhibit the most and largest structural holes, respectively. Since the owner is the core of the whole EPC network, it cannot be replaced, so it is concluded that the EPC contractor with the minimum aggregate constraint value is the critical agent.

Through application of the structural holes theory, it is found that there exists no direct connection or discontinuous relationship between the agents occurring in the network of industrial projects. Based on the whole network, it seems that there are holes in the network structure, namely, structural holes. The third agent (the EPC contractor) connecting two agents without a direct connection attains information and control advantages. Therefore, the EPC agent network of industrial projects should strive to occupy the third agent position among the structural holes, namely, the position of the EPC contractor.

**Table 1. Agent list of the EPC model.**

| Agent Name | Node Code | Node Description |
|---|---|---|
| Owner | A | |
| Government | B | |
| Financial Institution | C | |
| Consultant Agent | D | |
| Insurance Company | E | |
| Supervising Company 1 | F1 | Gasification, conversion, purification, pressure swing adsorption (PSA) hydrogen production |
| Supervising Company 2 | F2 | Methanol, sulfur recovery, formaldehyde, glycol |
| Supervising Company 3 | F3 | Air separation, coal storage and transportation, boiler operation |
| Supervising Company 4 | F4 | Auxiliary facilities |
| Feasibility Study Company | G | |
| General Design Company | H | |
| Licenser 1 | I1 | Gasification |
| Licenser 2 | I2 | Conversion |
| Licenser 3 | I3 | Purification |
| Licenser 4 | I4 | PSA hydrogen production |
| Licenser 5 | I5 | Sulfur recovery |
| Licenser 6 | I6 | Methanol |
| Licenser 7 | I7 | Formaldehyde |
| Licenser 8 | I8 | Glycol |
| Licenser 9 | I9 | Air separation |
| Licenser 10 | I10 | Boiler operation |
| FEED/Basic Design Company 1 | L1 | Gasification, conversion, purification, PSA hydrogen production |
| FEED/Basic Design Company 2 | L2 | Methanol, sulfur recovery, formaldehyde, glycol |
| FEED/Basic Design Company 3 | L3 | Air separation, coal storage and transportation, boiler operation |
| FEED/Basic Design Company 4 | L4 | Auxiliary facilities |
| EPC Contractor 1 | J1 | Gasification, conversion |
| EPC Contractor 2 | J2 | Purification, PSA hydrogen production |
| EPC Contractor 3 | J3 | Methanol, sulfur recovery, formaldehyde, glycol |
| EPC Contractor 4 | J4 | Formaldehyde, glycol |
| EPC Contractor 5 | J5 | Air separation |
| EPC Contractor 6 | J6 | Coal storage and transportation, boiler operation |
| EPC Contractor 7 | J7 | Auxiliary facilities |
| ⋮ | ⋮ | ⋮ |
| Commissioning Company 1 | K1 | Gasification, conversion, purification, PSA hydrogen production |
| Commissioning Company 2 | K2 | Methanol, sulfur recovery, formaldehyde, glycol |
| Commissioning Company 3 | K3 | Boiler operation |
| Commissioning Company 4 | K4 | Air separation |

Note: The design companies, suppliers and construction subcontractors of the EPC contractors are not reflected in detail in this table because there are too many agents.

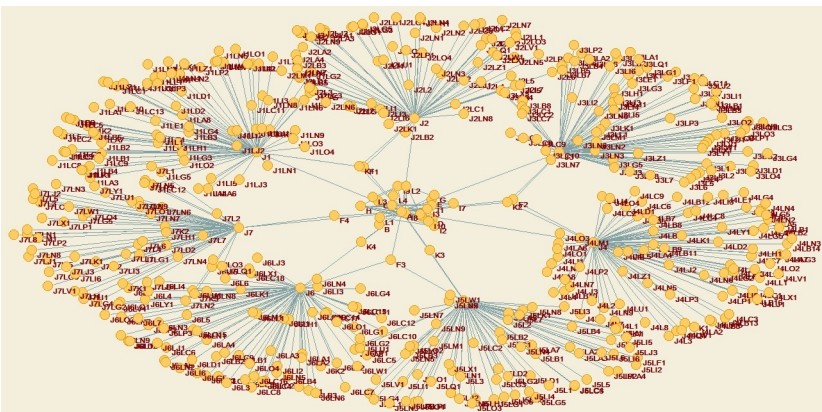

**Fig 6. Agent network under the EPC model.**

## 6.2 Complex network analysis of the engineering and construction multi-agents of major industrial projects under the PMC Model

There are a total of 541 agents under the PMC model, as shown in Table 4.

**6.2.1 Structural hole analysis of the engineering and construction multi-agents of major industrial projects under the PMC Model.** The agent network under the PMC model is shown in Fig 10.

In structural hole analysis, values $a_{ij}$ are assigned to the edges between nodes $i$ and $j$. Similarly, the agents under the PMC model are linked through contract relations, and the most critical contract index between the agents in the contract amount. Therefore, the contract amount is adopted as the attribute value of the edges between agents to construct a complex network, as shown in Table 5.

Through structural hole analysis under the PMC model, the aggregate constraint list is obtained, as shown in Table 6, a 2D network diagram is shown in Fig 11, a 3D network diagram is shown in Fig 12, and the 3D network acquired via VOS mapping is shown in Fig 13.

According to the aggregate constraint value, the owner and project management company attain the minimum aggregate constraints. Moreover, the 3D diagram of the aggregate constraint analysis of the structural holes shows that the owner and project management company contain the most and largest structural holes, respectively. Since the owner is the core of the whole PMC network, it cannot be replaced, and it is thus concluded that the project management company with the minimum aggregate constraint value is the critical agent.

Through application of the structural holes theory, it is found that the PMC agent network of industrial projects should strive to occupy the third agent position among the structural holes, namely, the position of the project management company.

## 7 Coordination mechanism of the major industrial project engineering and construction multi-agents

According to the above analysis of the complex network, it is concluded that under the different models, the engineering and construction agents compete for the dominant position among the structural holes because that position attains the dominant advantage. Therefore, to achieve the integration of multi-agent interests and coordination of multi-agent relationships,

**Table 2. List of the edge attribute values between the agents under the EPC model.**

| $i$ | $j$ | $a_{ij}$ |
|---|---|---|
| A | B | 100 |
| A | C | 100 |
| A | D | 1000 |
| A | E | 3000 |
| A | F1 | 500 |
| A | F2 | 400 |
| A | F3 | 300 |
| A | F4 | 250 |
| A | G | 150 |
| A | H | 1000 |
| A | I1 | 1000 |
| A | I2 | 100 |
| A | I3 | 300 |
| A | I4 | 100 |
| A | I5 | 200 |
| A | I6 | 800 |
| A | I7 | 500 |
| A | I8 | 2000 |
| A | I9 | 200 |
| A | I10 | 150 |
| A | L1 | 3000 |
| A | L2 | 5000 |
| A | L3 | 1000 |
| A | L4 | 800 |
| A | J1 | 222695 |
| A | J2 | 54138 |
| A | J3 | 87304 |
| A | J4 | 284500 |
| A | J5 | 40878 |
| A | J6 | 39864 |
| A | J7 | 25230 |
| F1 | J1 | 100 |
| F1 | J2 | 100 |
| F2 | J3 | 100 |
| F2 | J4 | 100 |
| F3 | J5 | 100 |
| F3 | J6 | 100 |
| F4 | J7 | 100 |
| A | K1 | 800 |
| A | K2 | 1200 |
| A | K3 | 500 |
| A | K4 | 400 |
| J1 | K1 | 100 |
| J2 | K1 | 100 |
| J3 | K2 | 100 |
| J4 | K2 | 100 |
| J5 | K3 | 100 |

(*Continued*)

**Table 2.** (Continued)

| i | j | $a_{ij}$ |
|---|---|---|
| J6 | K4 | 100 |
| H | I1 | 100 |
| H | I2 | 100 |
| H | I3 | 100 |
| H | I4 | 100 |
| H | I5 | 100 |
| H | I6 | 100 |
| H | I7 | 100 |
| H | I8 | 100 |
| H | I9 | 100 |
| H | I10 | 100 |
| I1 | L1 | 100 |
| I2 | L1 | 100 |
| I3 | L1 | 100 |
| I4 | L1 | 100 |
| I5 | L2 | 100 |
| I6 | L2 | 100 |
| I7 | L2 | 100 |
| I8 | L2 | 100 |
| I9 | L3 | 100 |
| I10 | L4 | 100 |
| ⋮ | ⋮ | ⋮ |

it is necessary to reduce the structural holes, whereas the aggregation coefficient of each agent tends to remain the same.

The aggregate constraint of the structural holes of each agent is relatively uniform in the PMC agent network, so the critical agent position of the project management company during project execution is not very obvious, not as obvious as the critical agent position of the EPC contractor in the EPC agent network. Therefore, the PMC model is more inclined to integrate multi-agent interests and coordinate multi-agent relationships.

According to the conducted structural hole analysis of the EPC and PMC models, it is concluded that when the aggregation coefficient of each agent tends to remain the same, the relationships among all agents and between the agent location in the network and that in the engineering and construction network tend to be more similar, which is conducive to the integration of multi-agent interests and coordination of multi-agent relationships, in addition to the stable development of the construction market.

## 8 Discussion

This study contributes to the literature by exploratively examining the coordination mechanism of the major industrial project engineering and construction multi-agents. There has been limited research into multi-agents relationship. The influence of organizational characteristics on agent and project performance is a direction of the multi-agents relationship, and how to establish an effective multi-agents social networks is another direction of multi-agents relationship. The prospect of this research is that it can be applied to large, complex and systematic industrial project management, and can be used as a guide for selecting project management model.

**Table 3. Aggregate constraint list of the agents under the EPC model.**

| Node Code | Aggregate Constraint |
| --- | --- |
| A | 0.239107 |
| B | 1 |
| C | 1 |
| D | 1 |
| E | 1 |
| F1 | 0.891819 |
| F2 | 0.920579 |
| F3 | 0.746002 |
| F4 | 0.829588 |
| G | 1 |
| H | 0.661507 |
| I1 | 0.914321 |
| I2 | 0.853987 |
| I3 | 0.849438 |
| I4 | 0.853987 |
| I5 | 0.860496 |
| I6 | 0.90974 |
| I7 | 0.885114 |
| I8 | 0.953296 |
| I9 | 0.852011 |
| I10 | 0.845111 |
| L1 | 0.89492 |
| L2 | 0.962055 |
| L3 | 0.919464 |
| L4 | 0.889431 |
| J1 | 0.347874 |
| J2 | 0.325535 |
| J3 | 0.269459 |
| J4 | 0.293377 |
| J5 | 0.283277 |
| J6 | 0.285006 |
| J7 | 0.270003 |
| K1 | 0.941948 |
| K2 | 1.037853 |
| K3 | 0.884161 |
| K4 | 0.867589 |

## 9 Conclusion

Industry is an important pillar of the national economy, and industrial projects are the most complex and difficult to manage and control in the construction industry; thus, the multi-agents coordination of industrial projects is one of the core issues for industrial construction projects. A large number of agents occurs in industrial projects, which constitute a complex agent network that cannot be analyzed with conventional methods.

In this paper, the agent network of industrial projects is constructed by analyzing the relationship of the engineering and construction multi-agent chains of industrial projects. On the basis of the agent network structure of industrial projects, the nonrepeated relationships

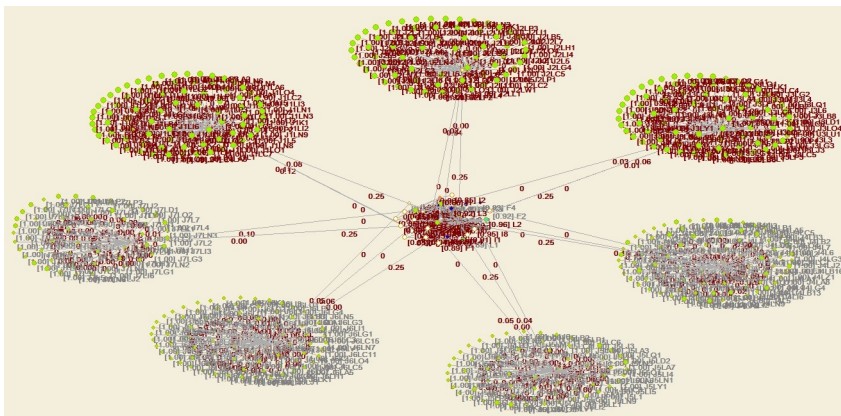

**Fig 7. Agent network and structural holes under the EPC model—2D diagram of aggregation constraint analysis.**

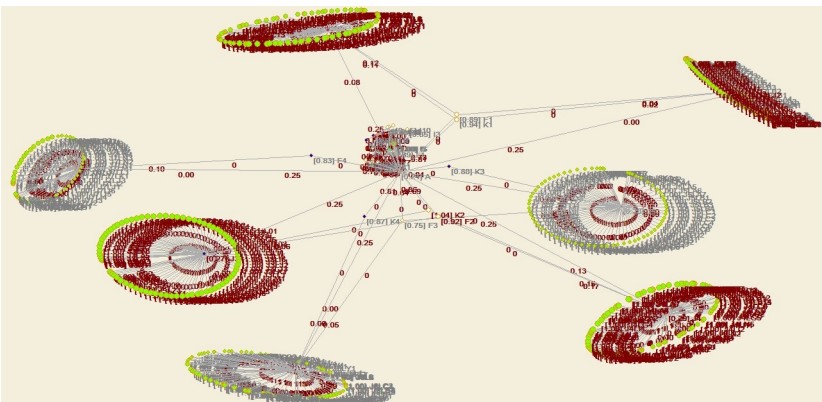

**Fig 8. Agent network and structural holes under the EPC model—3D diagram of aggregation constraint analysis.**

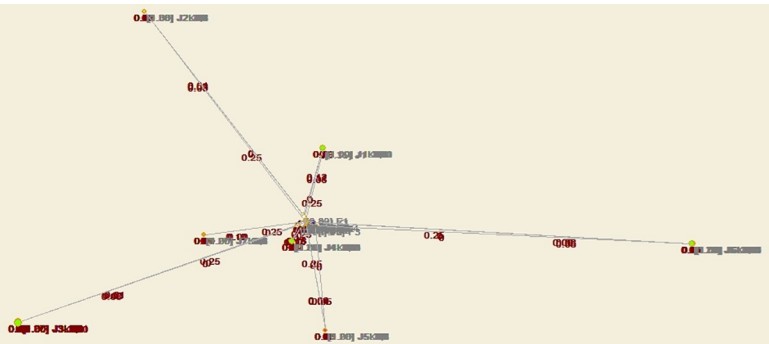

**Fig 9. Agent network and structural holes under the EPC model—VOS map of aggregation constraint analysis.**

between the network agents under the different models are analyzed via the structural holes theory. It is concluded that the characteristics of the agent maintain an advantage in competition, the coordination mechanism of the integration of agent interests, and multi-agent relations are considered to solve the multi-agent coordination problem in major industrial projects.

**Table 4. Agent list of the PMC model.**

| Agent Name | Node Code | Node Description |
|---|---|---|
| Owner | A | |
| Government | B | |
| Financial Institution | C | |
| Consultant Agent | D | |
| Insurance Company | E | |
| Supervising Company 1 | F1 | Gasification, conversion, purification, PSA hydrogen production |
| Supervising Company 2 | F2 | Methanol, sulfur recovery, formaldehyde, glycol |
| Supervising Company 3 | F3 | Air separation, coal storage and transportation, boiler operation |
| Supervising Company 4 | F4 | Auxiliary facilities |
| Feasibility Study Company | G | |
| General Design Company | H | |
| Licenser 1 | I1 | Gasification |
| Licenser 2 | I2 | Conversion |
| Licenser 3 | I3 | Purification |
| Licenser 4 | I4 | PSA hydrogen production |
| Licenser 5 | I5 | Sulfur recovery |
| Licenser 6 | I6 | Methanol |
| Licenser 7 | I7 | Formaldehyde |
| Licenser 8 | I8 | Glycol |
| Licenser 9 | I9 | Air separation |
| Licenser 10 | I10 | Boiler operation |
| FEED/Basic Design Company 1 | L1 | Gasification, conversion, purification, PSA hydrogen production |
| FEED/Basic Design Company 2 | L2 | Methanol, sulfur recovery, formaldehyde, glycol |
| FEED/Basic Design Company 3 | L3 | Air separation, coal storage and transportation, boiler |
| FEED/Basic Design Company 4 | L4 | Auxiliary facilities |
| Project management company | M | |
| Detailed Design Company 1 | J1K1 | Gasification |
| Detailed Design Company 2 | J1K2 | Conversion |
| Detailed Design Company3 | J2K1 | Purification |
| Detailed Design Company 4 | J3K1 | Methanol, sulfur recovery |
| Detailed Design Company 5 | J4K1 | Formaldehyde, glycol |
| Detailed Design Company 6 | J5K1 | PSA hydrogen production |
| Detailed Design Company 7 | J6K1 | Coal storage and transportation |
| Detailed Design Company 8 | J6K2 | Boiler operation |
| Detailed Design Company 9 | J7K1 | Auxiliary facilities 1 |
| Detailed Design Company 10 | J7K2 | Auxiliary facilities 2 |
| Detailed Design Company 11 | J7K3 | Administrative region |
| ⋮ | ⋮ | ⋮ |
| Commissioning Company 1 | K1 | Gasification, conversion, purification, PSA hydrogen production |
| Commissioning Company 2 | K2 | Methanol, sulfur recovery, formaldehyde, glycol |
| Commissioning Company 3 | K3 | Boiler operation |
| Commissioning Company 4 | K4 | Air separation |

Note: The design companies, suppliers and construction subcontractors are not reflected in detail in this table because there are too many agents.

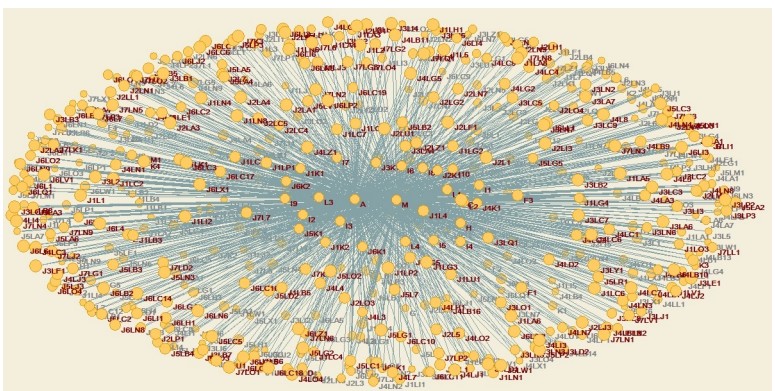

**Fig 10. Agent network under the PMC model.**

**Table 5. List of the edge attribute values between the agents under the PMC model.**

| i | j | $a_{ij}$ |
|---|---|---|
| A | B | 100 |
| A | C | 100 |
| A | D | 1000 |
| A | E | 3000 |
| A | F1 | 500 |
| A | F2 | 400 |
| A | F3 | 300 |
| A | F4 | 250 |
| A | G | 150 |
| A | H | 1000 |
| A | I1 | 1000 |
| A | I2 | 100 |
| A | I3 | 300 |
| A | I4 | 100 |
| A | I5 | 200 |
| A | I6 | 800 |
| A | I7 | 500 |
| A | I8 | 2000 |
| A | I9 | 200 |
| A | I10 | 150 |
| A | L1 | 3000 |
| A | L2 | 5000 |
| A | L3 | 1000 |
| A | L4 | 800 |
| A | M | 10000 |
| H | I1 | 100 |
| H | I2 | 100 |
| H | I3 | 100 |
| H | I4 | 100 |
| H | I5 | 100 |
| H | I6 | 100 |

*(Continued)*

**Table 5.** (Continued)

| i | j | $a_{ij}$ |
|---|---|---|
| H | I7 | 100 |
| H | I8 | 100 |
| H | I9 | 100 |
| H | I10 | 100 |
| I1 | L1 | 100 |
| I2 | L1 | 100 |
| I3 | L1 | 100 |
| I4 | L1 | 100 |
| I5 | L2 | 100 |
| I6 | L2 | 100 |
| I7 | L2 | 100 |
| I8 | L2 | 100 |
| I9 | L3 | 100 |
| I10 | L4 | 100 |
| L1 | J1K1 | 100 |
| L1 | J1K2 | 100 |
| L1 | J2K1 | 100 |
| L2 | J3K1 | 100 |
| L2 | J4K1 | 100 |
| L3 | J5K1 | 100 |
| L3 | J6K1 | 100 |
| L3 | J6K2 | 100 |
| A | J1K1 | 5000 |
| A | J1K2 | 800 |
| $\vdots$ | $\vdots$ | $\vdots$ |

**Table 6. Aggregate constraint list of the agents under the PMC model.**

| Node Code | Aggregate Constraint |
|---|---|
| A | 0.060541 |
| B | 0.590764 |
| C | 0.590764 |
| D | 0.863014 |
| E | 0.948306 |
| F1 | 0.770015 |
| F2 | 0.735227 |
| F3 | 0.690072 |
| F4 | 0.66298 |
| G | 0.605193 |
| H | 0.569726 |
| I1 | 0.793005 |
| I2 | 0.566507 |
| I3 | 0.634142 |
| I4 | 0.566507 |
| I5 | 0.621823 |
| I6 | 0.776284 |

(*Continued*)

**Table 6.** (Continued)

| Node Code | Aggregate Constraint |
|---|---|
| I7 | 0.714627 |
| I8 | 0.882314 |
| I9 | 0.568044 |
| I10 | 0.591354 |
| L1 | 0.83545 |
| L2 | 0.922496 |
| L3 | 0.764961 |
| L4 | 0.74836 |
| M | 0.495381 |
| J1K1 | 0.961418 |
| J1K2 | 0.823678 |
| J2K1 | 0.867778 |
| J3K1 | 0.923292 |
| J4K1 | 0.951105 |
| J5K1 | 0.830518 |
| J6K1 | 0.738826 |
| J6K2 | 0.763754 |
| J7K1 | 0.863014 |
| J7K2 | 0.863014 |
| J7K3 | 0.797158 |

This paper obtains the following research conclusions:

1. The agent position of the industrial project in the network is more important than the strength of its relationship. Its position in the network determines the information, resources and power of the agent. Therefore, if there is a structural hole, no matter how strong the agent relationship is, the third agent connecting any two agents without a direct contact attains both information and control advantages, which provides more services and returns for the third agent. Therefore, if the agent of an industrial project wants to maintain

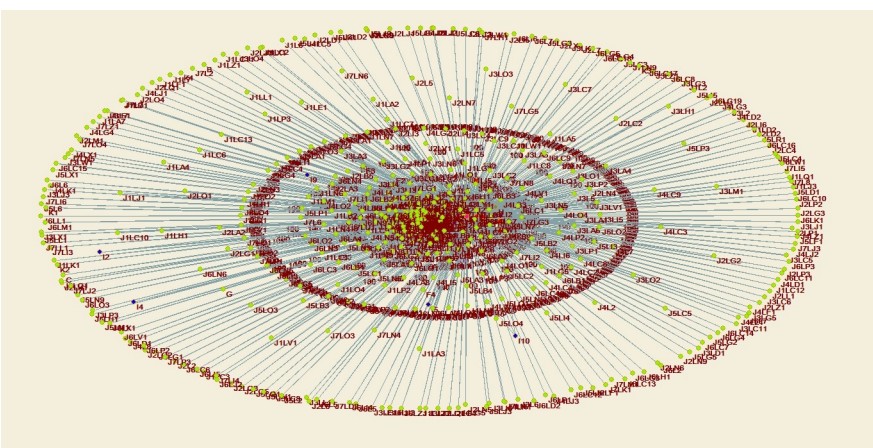

**Fig 11. Agent network and structural holes under the PMC model—2D diagram of aggregation constraint analysis.**

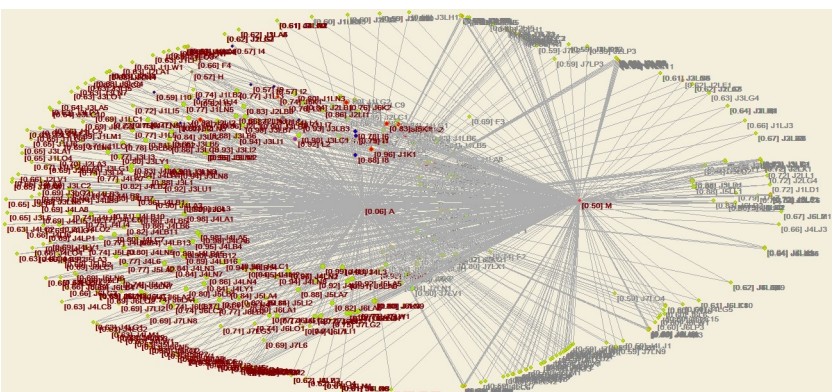

**Fig 12. Agent network and structural holes under the PMC model—3D diagram of aggregation constraint analysis.**

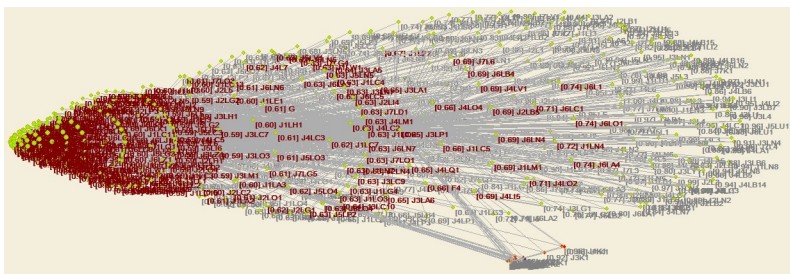

**Fig 13. Agent network and structural holes under the PMC model—VOS map of aggregation constraint analysis.**

an advantage in agent competition, it must establish a wide range of connections and occupy more structural holes.

2. Through network analysis of the EPC and PMC models, it is concluded that the PMC model facilitates the integration of multi-agent interests and coordination of multi-agent relationships, while the EPC model promotes the interests of the owner and EPC contractor.

3. When the aggregation coefficient of each agent tends to remain the same, the relationship between each agent and the agent location of the network in the engineering and construction network tends to be more equal, which is conducive to the integration of multi-agent interests and coordination of multi-agent relationships.

## Supporting information

**S1 Table. List of the edge attribute values between the agents under the EPC model.** (DOCX)

**S2 Table. List of the edge attribute values between the agents under the PMC model.** (DOCX)

## Acknowledgments

The authors thank the Editor, Associate Editor and referees for their comments on the initial version of the manuscript.

## Author Contributions

**Conceptualization:** Xiaokang Han.

**Data curation:** Xiaokang Han.

**Formal analysis:** Xiaokang Han.

**Methodology:** Xiaokang Han.

**Software:** Xiaokang Han.

**Supervision:** Wenzhou Yan, Mei Lu.

**Visualization:** Xiaokang Han.

**Writing – original draft:** Xiaokang Han.

**Writing – review & editing:** Xiaokang Han.

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
