## [Decision Letter · Decision Letter 0]

17 Jun 2021

PONE-D-21-16612

Research on the coordination mechanism of major industrial project engineering and construction multi-agents based on structural holes

PLOS ONE

Dear Dr. Han,

Thank you for submitting your manuscript to PLOS ONE. After careful consideration, we feel that it has merit but does not fully meet PLOS ONE’s publication criteria as it currently stands. Therefore, we invite you to submit a revised version of the manuscript that addresses the points raised during the review process.

Please consider all the comments of all reviewers including reviewer 2.

We look forward to receiving your revised manuscript.

Kind regards,

Ahmed Mancy Mosa, Ph.D.

Academic Editor

PLOS ONE

Journal Requirements:

"NO authors have competing interests."

We note that one or more of the authors are employed by a commercial company: Project Control Department, Hualu Engineering & Technology Co., Ltd.

2.1. Please provide an amended Funding Statement declaring this commercial affiliation, as well as a statement regarding the Role of Funders in your study. If the funding organization did not play a role in the study design, data collection and analysis, decision to publish, or preparation of the manuscript and only provided financial support in the form of authors' salaries and/or research materials, please review your statements relating to the author contributions, and ensure you have specifically and accurately indicated the role(s) that these authors had in your study. You can update author roles in the Author Contributions section of the online submission form.

2.2. Please also provide an updated Competing Interests Statement declaring this commercial affiliation along with any other relevant declarations relating to employment, consultancy, patents, products in development, or marketed products, etc.  

Reviewers' comments:

Reviewer's Responses to Questions

**Comments to the Author**

1. Is the manuscript technically sound, and do the data support the conclusions?

Reviewer #1: Yes

Reviewer #2: Partly

Reviewer #3: No

2. Has the statistical analysis been performed appropriately and rigorously? 

Reviewer #1: Yes

Reviewer #2: N/A

Reviewer #3: Yes

3. Have the authors made all data underlying the findings in their manuscript fully available?

Reviewer #1: Yes

Reviewer #2: Yes

Reviewer #3: Yes

4. Is the manuscript presented in an intelligible fashion and written in standard English?

Reviewer #1: Yes

Reviewer #2: Yes

Reviewer #3: No

5. Review Comments to the Author

Reviewer #1: Dear authors,

Thank you for submitting your paper to the PLOS ONE Journal. I believe your paper is an informative paper that can be published after a major revision. Please address my comments in the attached document.

Reviewer #2: Title: Research on the coordination mechanism of major industrial project engineering and construction multi-agents based on structural holes

This topic seems to be interesting and practical. The manuscript cannot be considered as a potential publication in its current form. However, there are some concerns to be resolved carefully.

1- The English and writing of the manuscript should improve. I found several errors.

2- Abstract should contain the main contributions and achievements of the study, particularly in a numerical way.

3- You should provide a useful survey on the related work addressing the main challenges, limitations, etc.

4- The information of all indices should be given in each mathematical equation.

5- What are the pros and cons of your proposed methodology?

6- Figures cannot be read and understood.

7- More comparisons should be made against the rival methods.

8- The outlook of the research is missed.

Reviewer #3: 1. Research keywords are not carefully selected. It is also necessary to give the reader an adequate explanation after choosing a keyword.

2. What was the lack of available knowledge that led to this research? Research novelty should be clearly stated in the abstract and introduction.

3. It is better to compare the results of this study with other similar studies. Also, the benefits of research are described.

4. In conclusion, the direction of research by other researchers in the future is not specified.

6. PLOS authors have the option to publish the peer review history of their article (what does this mean?). If published, this will include your full peer review and any attached files.

Reviewer #1: No

Reviewer #2: No

Reviewer #3: **Yes: **Ali Reza Afshari

---

## [Author Response · Author response to Decision Letter 0]

23 Jun 2021

See the attachment for Response to Reviewers.

Below are the responses to the comments:

Reviewer #1: 

Thank you for submitting your paper to the Journal of PLOS ONE. I believe your paper is an informative paper which can be published after a major revision.

Comments:

1. It is not clear what the research gap that the paper is addressing. What is the objective of this paper? Please clarify somewhere clearly all your contributions. 

Reply:

The agent coordination problem of industrial engineering and construction projects is difficult to solve, which requires a new analysis method of the essential characteristics. This paper attempts to solve the multi-agent coordination problem in major industrial projects.

This study contributes to the literature by exploratively examining the coordination mechanism of the major industrial project engineering and construction multi-agents. There has been limited research into multi-agents relationship. The influence of organizational characteristics on agent and project performance is a direction of the multi-agents relationship, and how to establish an effective multi-agents social networks is another direction of multi-agents relationship.

2. The literature review is not goal oriented. The process should be as follows:

i) Critical evaluation of the literature; ii) identifying the gap based on this critical evaluation of the literature; iii) proposing your hypothesis to address the identified gap; iv) posing the appropriate and relevant research question based on your proposed hypothesis; and finally explaining your proposed method to answer this research question. Therefore, you will have a systematic way of conducting your research. Right now, the literature review section has no clear objective. 

Reply: It has been revised.

3. An adequate literature review and a clear gap identification have been tried to be conducted. However, authors have ignored some research which has been done in the area. I strongly recommend the authors to provide a more comprehensive literature review in the introduction section. The following papers are recommended:

• Social cohesion, structural holes, and a tale of two measures. Journal of Statistical Physics, 151(3), 745-764.

• Knowledge, networks, and knowledge networks: A review and research agenda. Journal of management, 38(4), 1115-1166.

• Performance evaluation of complex electricity generation systems: A dynamic network-based data envelopment analysis approach. Energy Economics, 104894.

• Complementary or contradictory? The effects of structural holes and status on innovation. Innovation, 20(4), 393-406.

Reply: The references have been added.

In 2012, Phelps et al.[9] developed a comprehensive framework that organizes the knowledge networks literature, which was used to review extant empirical research within and across multiple disciplines and levels of analysis, and identified points of coherence and conflict in theoretical arguments and empirical results within and across levels and identified emerging themes and promising areas for future research. In 2015, Burt [10] studied the strengthening of structural holes and proposed a reinforcement measure. In 2018, Tohyun et al. [11] studied focuses on the joint effects of the firms’ access to structural holes within social networks and their status within social hierarchy on their innovation performance, argued that the effects of structural holes and status contradict, rather than complement, each other because one tends to interfere with the other, and found that the positive effects of structural holes tend to be relatively stronger among lower-status firms, whereas the negative effects become stronger as the firms’ status increases. In 2019, Deng et al. [12] proposed a new measurement model for critical nodes based on global features and local features, which considers the edge betweenness and edge clustering coefficients and combines the mutual influence between nodes and edges in a network, and proposed an algorithm based on the aforementioned model Subsequently. In 2020, Alizadeh et al. [13] proposed a dynamic DEA (DDEA) model with time-based dependencies between the successive periods for assessing the performance over successive periods, and found that the efficiencies of power generation and transmission sectors are decreasing while the distribution performance is increasing.

4. A thorough editorial check and English improvement are needed. Please kindly proofread the entire manuscript.

Reply: It has been revised.

5. The conclusion part is also needed to be revised; which questions are answered, what is the value/originality/contribution of the paper, how the proposed method answers the research questions that previous methods are not able to answer? 

Reply: It has been revised.

Industry is an important pillar of the national economy, and industrial projects are the most complex and difficult to manage and control in the construction industry; thus, the multi-agents coordination of industrial projects is one of the core issues for industrial construction projects.

A large number of agents occurs in industrial projects, which constitute a complex agent network that cannot be analyzed with conventional methods. Therefore, it is necessary to analyze the characteristics of the topological structure of the complex network of the agents occurring in industrial projects through software suitable for the analysis and visualization of large networks with thousands or millions of nodes.

In this paper, the agent network of industrial projects is constructed by analyzing the relationship of the engineering and construction multi-agent chains of industrial projects. On the basis of the agent network structure of industrial projects, the nonrepeated relationships between the network agents under the different models are analyzed via the structural holes theory. It is concluded that the characteristics of the agent maintain an advantage in competition, the coordination mechanism of the integration of agent interests, and multi-agent relations are considered to solve the multi-agent coordination problem in major industrial projects.

6. It feels you need a king of aggregating results somewhere clearer. 

Reply: It has been revised.

According to the above analysis of the complex network, it is concluded that under the different models, the engineering and construction agents compete for the dominant position among the structural holes because that position attains the dominant advantage. Therefore, to achieve the integration of multi-agent interests and coordination of multi-agent relationships, it is necessary to reduce the structural holes, whereas the aggregation coefficient of each agent tends to remain the same.

According to the conducted structural hole analysis of the EPC and PMC models, it is concluded that when the aggregation coefficient of each agent tends to remain the same, the relationships among all agents and between the agent location in the network and that in the engineering and construction network tend to be more similar, which is conducive to the integration of multi-agent interests and coordination of multi-agent relationships, in addition to the stable development of the construction market.

7. Please propose and suggest more possible future studies related to the current study. 

Reply: Discussion has been added.

This study contributes to the literature by exploratively examining the coordination mechanism of the major industrial project engineering and construction multi-agents. There has been limited research into multi-agents relationship. The influence of organizational characteristics on agent and project performance is a direction of the multi-agents relationship, and how to establish an effective multi-agents social networks is another direction of multi-agents relationship.

8. If you can, please make a small comparison between what did you do and what others did before; as a conclusion.

Reply: The research on the coordination mechanism of the major industrial project engineering and construction multi-agents is sparse.

9. The abstract is not deep enough and Is not well prepared. Please try to re-write it better. The problem should be clearly stated and the gap which you are going to address need to be clarified. Simply explain your contributions and key findings.

Reply: The abstract has been revised.

Industry is an important pillar of the national economy. Industrial projects are the most complex and difficult projects to control in the construction industry, and major industrial projects are even more complex and difficult to control. Multi-agent coordination is one of the core issues of industrial projects. Based on an analysis of the engineering and construction chains and agent relationships and agent networks of industrial projects, a complex network of the engineering and construction agents of industrial projects is established, and the complex network structural holes theory is applied to study the nonrepeated relationships among agents in industrial projects. Assuming agents are linked through contract relations and the most critical contract index between the agents in the contract amount, through structural hole analysis considering the EPC and PMC model, the aggregate constraint list is obtained, 2D network diagram and 3D network diagram are shown. According to the aggregate constraint value, the EPC contractor with the minimum aggregate constraint value and the project management company with the minimum aggregate constraint value are the critical agent in EPC and PMC model. By analyzing the complex network comprising different models of industrial projects, it is concluded that the characteristics of the agent maintain an advantage in competition, the coordination mechanism of the integration of agent interests, and multi-agent relations are considered to solve the multi-agent coordination problem in major industrial projects.

10. There are some errors in your reference list. Please check and fix the errors.

Reply: It has been modified.

Reviewer #2:

Title: Research on the coordination mechanism of major industrial project engineering and construction multi-agents based on structural holes

This topic seems to be interesting and practical. The manuscript cannot be considered as a potential publication in its current form. However, there are some concerns to be resolved carefully.

1. The English and writing of the manuscript should improve. I found several errors.

Reply: It has been modified.

2. Abstract should contain the main contributions and achievements of the study, particularly in a numerical way.

Reply: The abstract has been revised.

Industry is an important pillar of the national economy. Industrial projects are the most complex and difficult projects to control in the construction industry, and major industrial projects are even more complex and difficult to control. Multi-agent coordination is one of the core issues of industrial projects. Based on an analysis of the engineering and construction chains and agent relationships and agent networks of industrial projects, a complex network of the engineering and construction agents of industrial projects is established, and the complex network structural holes theory is applied to study the nonrepeated relationships among agents in industrial projects. Assuming agents are linked through contract relations and the most critical contract index between the agents in the contract amount, through structural hole analysis considering the EPC and PMC model, the aggregate constraint list is obtained, 2D network diagram and 3D network diagram are shown. According to the aggregate constraint value, the EPC contractor with the minimum aggregate constraint value and the project management company with the minimum aggregate constraint value are the critical agent in EPC and PMC model. By analyzing the complex network comprising different models of industrial projects, it is concluded that the characteristics of the agent maintain an advantage in competition, the coordination mechanism of the integration of agent interests, and multi-agent relations are considered to solve the multi-agent coordination problem in major industrial projects.

3. You should provide a useful survey on the related work addressing the main challenges, limitations, etc.

Reply: Discussion has been added.

This study contributes to the literature by exploratively examining the coordination mechanism of the major industrial project engineering and construction multi-agents. There has been limited research into multi-agents relationship. The influence of organizational characteristics on agent and project performance is a direction of the multi-agents relationship, and how to establish an effective multi-agents social networks is another direction of multi-agents relationship.

4. The information of all indices should be given in each mathematical equation.

Reply: The information of all indices is as follows:

A structural hole is considered to measure the constraint index of networks , which is the ratio of the probability value of the relation between nodes i and j to the probability value of all relations of i, and is the value of the edge attribute between nodes i and j.

 (1)

Aggregate constraints denote the missing constraints of node j around the initial hole of node i.

 (2)

K is the set of all nodes connected to node i, for .

The higher the coefficient, the fewer structural holes there are and the higher the network closure is.

The aggregate constraint of node i is , corresponding to independent nodes =1.

 (3)

5. What are the pros and cons of your proposed methodology?

Reply: 

The pros is that it emphasizes that structural holes in interpersonal networks can bring advantages in information and other resources to organizations and individuals in that position. If there is no direct connection between the two, and the connection must be formed through the third party, then the third party acting occupies a structural hole in the network of the relationship. Obviously, the structural hole is for the third party.

The cons is that if there is no third party, or the network structure is not complex, it is difficult to reflect the advantages of structural hole theory.

6. Figures cannot be read and understood.

Reply: The figures are the visualization of 547 agents under the EPC model and 541 agents under the PMC model by Pajek software, so they look complex.

7. More comparisons should be made against the rival methods.

Reply: The related references have been added.

8. The outlook of the research is missed.

Reply: Discussion has been added.

The prospect of this research is that it can be applied to large, complex and systematic industrial project management, and can be used as a guide for selecting project management model.

Reviewer #3: 

1. Research keywords are not carefully selected. It is also necessary to give the reader an adequate explanation after choosing a keyword.

Reply: The key words have been revised; the key words are industrial project; major industrial project; multi-agents; coordination mechanism; complex network; structural holes. Because this paper is to study the multi-agents relationship of major industrial projects, and apply the structural hole theory.

2. What was the lack of available knowledge that led to this research? Research novelty should be clearly stated in the abstract and introduction.

Reply: With the development of science and technology, industrial projects tend to be increasingly large-scale and intensive projects. Industrial projects are characterized by a large scale, high investment, complex process system, intensive layout, open-air features, automatic control, high construction technology requirements, numerous design institutes, several suppliers, many construction units, various cooperation units, countless work interface relations, and long construction cycles, and the agent coordination problem of industrial engineering and construction projects is difficult to solve, which requires a new analysis method of the essential characteristics.

The abstract and introduction have been revised.

3. It is better to compare the results of this study with other similar studies. Also, the benefits of research are described.

Reply: The research on the coordination mechanism of the major industrial project engineering and construction multi-agents is sparse.

The advantages are that it emphasizes that structural holes in interpersonal networks can bring advantages in information and other resources to organizations and individuals in that position. If there is no direct connection between the two, and the connection must be formed through the third party, then the third party acting occupies a structural hole in the network of the relationship. Obviously, the structural hole is for the third party.

4. In conclusion, the direction of research by other researchers in the future is not specified.

Reply: Discussion has been added.

This study contributes to the literature by exploratively examining the coordination mechanism of the major industrial project engineering and construction multi-agents. There has been limited research into multi-agents relationship. The influence of organizational characteristics on agent and project performance is a direction of the multi-agents relationship, and how to establish an effective multi-agents social networks is another direction of multi-agents relationship. The prospect of this research is that it can be applied to large, complex and systematic industrial project management, and can be used as a guide for selecting project management model.

---

## [Decision Letter · Decision Letter 1]

7 Jul 2021

PONE-D-21-16612R1

Research on the coordination mechanism of major industrial project engineering and construction multi-agents based on structural holes

PLOS ONE

Dear Dr. Han,

Thank you for submitting your manuscript to PLOS ONE. After careful consideration, we feel that it has merit but does not fully meet PLOS ONE’s publication criteria as it currently stands. Therefore, we invite you to submit a revised version of the manuscript that addresses the points raised during the review process.

Please consider the comments of reviewer 3

We look forward to receiving your revised manuscript.

Kind regards,

Ahmed Mancy Mosa, Ph.D.

Academic Editor

PLOS ONE

Journal Requirements:

Reviewers' comments:

Reviewer's Responses to Questions

**Comments to the Author**

1. If the authors have adequately addressed your comments raised in a previous round of review and you feel that this manuscript is now acceptable for publication, you may indicate that here to bypass the “Comments to the Author” section, enter your conflict of interest statement in the “Confidential to Editor” section, and submit your "Accept" recommendation.

Reviewer #1: All comments have been addressed

Reviewer #2: All comments have been addressed

Reviewer #3: (No Response)

2. Is the manuscript technically sound, and do the data support the conclusions?

Reviewer #1: Yes

Reviewer #2: Yes

Reviewer #3: No

3. Has the statistical analysis been performed appropriately and rigorously? 

Reviewer #1: Yes

Reviewer #2: Yes

Reviewer #3: No

4. Have the authors made all data underlying the findings in their manuscript fully available?

Reviewer #1: Yes

Reviewer #2: Yes

Reviewer #3: No

5. Is the manuscript presented in an intelligible fashion and written in standard English?

Reviewer #1: Yes

Reviewer #2: Yes

Reviewer #3: Yes

6. Review Comments to the Author

Reviewer #1: Dear authors,

Thank you for your efforts in revising the manuscript. I believe you did a great job in the revision and the changes alleviated my concerns regarding the manuscript. Therefore, I recommend its publication.

Looking forward to seeing the published version of your paper.

Good luck

Reviewer #2: The authors could address all the comments and improve the quality of the manuscript. It can be now accepted.

Reviewer #3: Unfortunately, this study does not have sufficient conditions for confirmation. This study still does not have sufficient conditions for confirmation. These confusing images, inadequate descriptions, inefficient sources, and unsystematic writing can lead to disapproval of the article.

7. PLOS authors have the option to publish the peer review history of their article (what does this mean?). If published, this will include your full peer review and any attached files.

Reviewer #1: No

Reviewer #2: No

Reviewer #3: **Yes: **Ali Reza Afshari

---

## [Author Response · Author response to Decision Letter 1]

14 Jul 2021

Here are the responses to the comments:

Reviewer #3:

1. Research keywords are not carefully selected. It is also necessary to give the reader an adequate explanation after choosing a keyword.

Reply: The key words have been revised. The key words are industrial project; major industrial project; multi-agents; coordination mechanism; complex network; structural holes theory. Because the paper is to study the mechanism of the multi-agents relationship of major industrial projects based on the structural holes theory, so the above keywords are selected.

2. What was the lack of available knowledge that led to this research? Research novelty should be clearly stated in the abstract and introduction.

Reply: With the development of science and technology, industrial projects tend to be increasingly large-scale and intensive projects. Industrial projects are characterized by a large scale, high investment, complex process system, intensive layout, open-air features, automatic control, high construction technology requirements, numerous design institutes, several suppliers, many construction units, various cooperation units, countless work interface relations, and long construction cycles, and the agent coordination problem of industrial engineering and construction projects is difficult to solve, which requires a new analysis method of the essential characteristics.

The abstract and introduction have been revised, as following:

Abstract

Industry is an important pillar of the national economy. Industrial projects are the most complex and difficult projects to control in the construction industry, and major industrial projects are even more complex and difficult to control. Multi-agent coordination is one of the core issues of industrial projects. Based on an analysis of the engineering and construction chains and agent relationships and agent networks of industrial projects, a complex network of the engineering and construction agents of industrial projects is established, and the complex network structural holes theory is applied to study the nonrepeated relationships among agents in industrial projects. Assuming agents are linked through contract relations and the most critical contract index between the agents in the contract amount, through structural hole analysis considering the EPC and PMC model, the aggregate constraint list is obtained, 2D network diagram and 3D network diagram are shown. According to the aggregate constraint value, the EPC contractor with the minimum aggregate constraint value and the project management company with the minimum aggregate constraint value are the critical agent in EPC and PMC model. By analyzing the complex network comprising different models of industrial projects, it is concluded that the characteristics of the agent maintain an advantage in competition, the coordination mechanism of the integration of agent interests, and multi-agent relations are considered to solve the multi-agent coordination problem in major industrial projects.

1 Introduction

Major projects include infrastructure or plant projects with a large investment scale, high technical complexity and difficult control process, such as transportation, energy, chemical industry, communications, municipal administration and aerospace projects, which exert an important influence on political, economic, social, science and technology, engineering development, environmental protection, public health, national security and strategic planning aspects.

With the development of science and technology, industrial projects tend to be increasingly large-scale and intensive projects. Industrial projects are characterized by a large scale, high investment, complex process system, intensive layout, open-air features, automatic control, high construction technology requirements, numerous design institutes, several suppliers, many construction units, various cooperation units, countless work interface relations, and long construction cycles, and the agent coordination problem of industrial engineering and construction projects is difficult to solve, which requires a new analysis method of the essential characteristics.

3. It is better to compare the results of this study with other similar studies. Also, the benefits of research are described.

Reply: The similar studies have been revised.

3 Structural holes theory

Structural holes are the nonrepeated relationships between two agents. The benefits contributed by the two agents associated with the structural holes to the network are accumulative but not overlapping. There are two indexes to measure structural holes: cohesion and structural allelism. If strong ties occur between two agents, repeated agent relations exist that lack structural holes, and cohesion redundancy follows. A structural hole exists if two of neighbours are not linked to each other. Through these two neighbours, they are connected to different parts of the larger network, and thus have access to different sources of dispersed information. Thus if a firm is to form a new link, closing a structural hole is less valuable than finding a partner to whom none of the current partners is currently connected [19]. The argument is that agents attempt to increase their betweenness centrality. Rodan empirically tested a mediated moderation model that distinguishes between the five different theoretical mechanisms: autonomy, competition, information brokering, opportunity recognition and innovativeness, and the findings suggested that of these five theoretical causal motors, innovativeness plays a key role in linking network structure and network content to performance [20]. Xing et al. analyzed the spreading effect of industrial sectors with complex network model under perspective of econophysics, and the industrial complex network based on input-output tables from WIOD was proposed to be a bridge connecting accurate static quantitative analysis and comparable dynamic one [21]. Latora et al. attempted to reconcile closed and open structures by proposing a new measure, Simmelian brokerage, that captures opportunities of brokerage between otherwise disconnected cohesive groups of contacts, and the implications of our findings for research on social capital and complex networks were discussed [22].

Zhao et al. proposed a novel measure based on Structural Holes and Degree Centrality(SHDC) which combined Structural Hole and Degree Centrality to measure the node influence, and the method used Degree Centrality to make a fast and coarse distinction between the influence of nodes and uses Structure Hole to reflect the impact of topological connections among neighbor nodes, which improved the ability to distinguish the influence of nodes in the low time complexity [23].

The advantages of structural holes are that it emphasizes that structural holes in interpersonal networks can bring advantages in information and other resources to organizations and individuals in that position. If there is no direct connection between the two, and the connection must be formed through the third party, then the third party acting occupies a structural hole in the network of the relationship. Obviously, the structural hole is for the third party.

References

1. Freeman LC. A Set of Measures of Centrality Based on Betweenness. Sociometry. 1977; 40(1):35-41. https://doi.org/10.2307/3033543

2. Freeman LC. Centrality in social networks : Conceptual clarification. Social Network. 1979; 1(3):215-239. https://doi.org/10.1016/0378-8733(78)90021-7#doilink

3. Freeman LC, Roeder D, Mulholland RR. Centrality in Social Networks: II. Experimental Results. Social Networks. 1980; 2(2):119-141. https://doi.org/10.1016/0378-8733(79)90002-9

4. Burt RS. Structural Holes: The Social Structure of Competition. Cambridge MA: Harvard University Press; 1992.

5. Watts DJ, Strogatz SH. Collective dynamics of 'small-world' networks. Nature. 1998; 393(6684):440-440.

6. Barabási AL, Albert R.Emergence of scaling in random networks. Science. 1999; 286(5499):509-512.

7. Rodan S. Structural holes and managerial performance: Identifying the underlying mechanisms. Social Networks. 2010; 32(3):168-179. https://doi.org/10.1016/j.socnet.2009.11.002

8. Latora V, Nicosia V, Panzarasa P. Social Cohesion, Structural Holes, and a Tale of Two Measures. Journal of Statistical Physics. 2012; 151(3):745-764. https://doi.org/10.1007/s10955-013-0722-z

9. Phelps C, Heidl R, Wadhwa A. Knowledge, Networks, and Knowledge Networks: A Review and Research Agenda. Journal of Management. 2012; 38(4):1115-1166. https://doi.org/10.1177/0149206311432640

10. Burt RS. Reinforced structural holes. Social Networks. 2015; 43:149-161. https://doi.org/10.1016/j.socnet.2015.04.008

11. Tohyun K, Kisang P, Eunjung K. Complementary or contradictory? The effects of structural holes and status on innovation. Innovation Management Policy & Practice, 2018; 20(4):393-406. https://doi.org/10.1080/14479338.2018.1478733

12. Deng YJ, Li YQ, Yin RR ,Zhao HY, Liu B. Efficient measurement model for critical nodes based on edge clustering coefficients and edge betweenness. Wireless Networks. 2019; 26(1): 2785-2795. https://doi.org/10.1007/s11276-019-02040-4

13. Alizadeh R, Beiragh RG, Soltanisehat L, Soltanzadeh E， Lund PD. Performance evaluation of complex electricity generation systems: A dynamic network-based data envelopment analysis approach. Energy Economics. 2020; 104894. https://doi.org/10.1016/j.eneco.2020.104894

14. Han XK, Yan WZ, Lu M. Research on reasoning concerning emergency measures for industrial project scheduling control. Advances in Civil Engineering. 2021; 5595354. https://doi.org/10.1155/2021/5595354

15. Han XK.WBS-free scheduling method based on database relational model. Int J Syst Assur Eng Manag. 2021; 01106-x. https://doi.org/10.1007/s13198-021-01106-x

16. Han XK, Yan WZ, Lu M. Intelligent Critical Path Computation Algorithm Utilising Ant Colony Optimisation for Complex Project Scheduling. Complexity 2021; 9930113. https://doi.org/10.1155/2021/9930113

17. Nooy WD, Mrvar A, Batagelj V. Exploratory Social Network Analysis with Pajek: Revised and Expanded Edition for Updated Software. Third Edition. New York: Cambridge University Press; 2018.

18. Mrvar A, Batagelj V. Program for Analysis and Visualization of Large Networks Reference Manual List of commands with short explanation version 5.10. Slovenia: University of Ljubljana; 2020.

19. Cowan R, Jonard N. Structural holes, innovation and the distribution of ideas. J Econ Interac Coord. 2007; 2: 93–110. https://doi.org/10.1007/s11403-007-0024-0

20. Rodan S. Structural holes and managerial performance: identifying the underlying mechanisms. Social Networks. 2010; 32(3): 168-179. https://doi.org/10.1016/j.socnet.2009.11.002

21. Xing L, Ye Q, Guan J, Adriana B L. Spreading Effect in Industrial Complex Network Based on Revised Structural Holes Theory. Plos One. 2016, 11(5): e0156270. https://doi.org/10.1371/journal.pone.0156270

22. Latora V, Nicosia V, Panzarasa P. Social Cohesion, Structural Holes, and a Tale of Two Measures. J Stat Phys. 2013, 151: 745–764. https://doi.org/10.1007/s10955-013-0722-z

23. Zhao X, Guo S, Wang Y. The Node Influence Analysis in Social Networks Based on Structural Holes and Degree Centrality. 2017 IEEE International Conference on Computational Science and Engineering (CSE) and IEEE International Conference on Embedded and Ubiquitous Computing (EUC). 2017, 708-711. https://doi.org/10.1109/CSE-EUC.2017.137.

4. In conclusion, the direction of research by other researchers in the future is not specified.

Reply: Discussion has been added.

This study contributes to the literature by exploratively examining the coordination mechanism of the major industrial project engineering and construction multi-agents. There has been limited research into multi-agents relationship. The influence of organizational characteristics on agent and project performance is a direction of the multi-agents relationship, and how to establish an effective multi-agents social networks is another direction of multi-agents relationship. The prospect of this research is that it can be applied to large, complex and systematic industrial project management, and can be used as a guide for selecting project management model.

---

## [Decision Letter · Decision Letter 2]

26 Jul 2021

Research on the coordination mechanism of major industrial project engineering and construction multi-agents based on structural holes theory

PONE-D-21-16612R2

Dear Dr. Han,

We’re pleased to inform you that your manuscript has been judged scientifically suitable for publication and will be formally accepted for publication once it meets all outstanding technical requirements.

Kind regards,

Ahmed Mancy Mosa, Ph.D.

Academic Editor

PLOS ONE

Additional Editor Comments (optional):

Reviewers' comments:

Reviewer's Responses to Questions

**Comments to the Author**

1. If the authors have adequately addressed your comments raised in a previous round of review and you feel that this manuscript is now acceptable for publication, you may indicate that here to bypass the “Comments to the Author” section, enter your conflict of interest statement in the “Confidential to Editor” section, and submit your "Accept" recommendation.

Reviewer #3: All comments have been addressed

2. Is the manuscript technically sound, and do the data support the conclusions?

Reviewer #3: No

3. Has the statistical analysis been performed appropriately and rigorously? 

Reviewer #3: I Don't Know

4. Have the authors made all data underlying the findings in their manuscript fully available?

Reviewer #3: Yes

5. Is the manuscript presented in an intelligible fashion and written in standard English?

Reviewer #3: Yes

6. Review Comments to the Author

Reviewer #3: Ok

This study contributes to the literature by exploratively examining the coordination mechanism of the major industrial project engineering and construction multi-agents. There has been limited research into multi-agents relationship.

7. PLOS authors have the option to publish the peer review history of their article (what does this mean?). If published, this will include your full peer review and any attached files.

Reviewer #3: **Yes: **Ali Reza Afshari

---

## [Editor Report · Acceptance letter]

28 Jul 2021

PONE-D-21-16612R2 

Research on the coordination mechanism of major industrial project engineering and construction multi-agents based on structural holes theory 

Dear Dr. Han:

I'm pleased to inform you that your manuscript has been deemed suitable for publication in PLOS ONE. Congratulations! Your manuscript is now with our production department. 

Kind regards, 

on behalf of

Dr. Ahmed Mancy Mosa 

Academic Editor

PLOS ONE